# Flash Flood Risk Perception by the Population of Mindelo, S. Vicente (Cape Verde)

**Bruno Martins** [1,*] , **Adélia Nunes** [2] , **Luciano Lourenço** [2] and **Fátima Velez-Castro** [2]

1   CEGOT (Centre of Studies on Geography and Spatial Planning), RISCOS, University of Coimbra, 3004-530 Coimbra, Portugal

2   Department of Geography and Tourism, CEGOT (Centre of Studies on Geography and Spatial Planning), University of Coimbra, 3004-530 Coimbra, Portugal; adeliajnnunes@gmail.com (A.N.); luciano@uc.pt (L.L.); velezcastro@fl.uc.pt (F.V.-C.)

*   Correspondence: bmscmartins@gmail.com

**Abstract:** São Vicente Island (Republic of Cape Verde) lies within the Sahelian zone and faces several natural hazards, one of which is flash flooding. With the purpose of understanding what factors determine flash flood risk perception, a questionnaire entitled Flash Flood Hazard Perception in Cape Verde was applied to 199 subjects. Analysis of variance (ANOVA) was performed to identify the primary factors associated with the perception of flash flood risk. Differences between different groups under the same impact factor were also compared. The results indicated that certain socio-demographic characteristics of the respondents (gender, level of education, and type of housing) and prior experience correlated with flash flood risk perception. The study also shows statistical differences between the groups. In general, males and the respondents with a high level of education, homeowners, and people with prior experience have better perception of the flash flood risk. These findings can help decision makers to improve effective flash flood risk communication policies and flood risk reduction strategies.

**Keywords:** risk perception; flash floods; São Vicente Island; Cape Verde

---

## 1. Introduction

The number of disasters related to natural hazards and their impact has significantly increased in recent decades [1–3]. The resulting economic and social costs, especially those related to losses/damages and the recovery/reconstruction processes are certainly substantial. Flood hazards in particular cost nearly 20,000 lives and affect 20 million people worldwide every year, mostly because of the resulting homelessness [4].

Even though flash floods were mainly studied in the context of physical science in the last century, in 1942, in a seminar about the human reaction to a flash flood crisis scenario ("Human Adjustment to Flood") White acknowledged the important value of human perception in the mitigation process, stressing the human factor as determinant in the risk perception (RP) [5] and risk communication [6]. According to Wachinger et al. (2013) [7] risk perception is the process of collecting, selecting, and interpreting signals about the uncertain impacts of events and involves multiple influencing factors in a very complex framework [8–12]. On the other hand, risk perception depends on an individual's subjective judgement and evaluation of a specific risk [13–15] which can be perceived as potentially dangerous by one person, whilst it may be considered safe by another person.

Risk management is, therefore, the modulated mental models and the psychological mechanisms that people use to judge, evaluate, tolerate, and react to risks [16]. Furthermore, it is how individuals and communities perceive the complex and varied factors which interfere in risk perception, such as

social networks and capital, media influence, personal experience, values, worldviews, and the influence of individual adaptation strategy through learning from past events [17–21].

Therefore, several studies have focused on the 'subjective' component of flood risk, and paying increased attention to the perception of flood risk has been recognized as a key element in its management, leading to an ongoing combination of social variables with more conventional risk estimation methods, mainly focusing on Europe and North America [22–30].

This study aims to determine what factors influence flash flood risk perception in Mindelo, São Vicente, in the archipelago of Cape Verde, where little-to-nothing is known about public perception of the risk posed by flash floods. More specifically, the objectives were to understand how people perceived the flash flood risk (with regard to the causal attributions, knowledge level, perception of support from public entities regarding flash floods) and the relationships between various factors that generally influence such perception, in particular the socio-economic and demographic characteristics of the population (gender, age, education, income), their housing status (owner or renter) and their prior experience of flash floods.

It is assumed that the acknowledgement and the understanding of these factors will help to improve decision makers' awareness as to the type of variables they must consider when conceptualizing efficient communication strategies to be contemplated in any plans for preventive measures to combat and remedy natural hazards. In fact, in this context it should be a priority to understand the characteristics of local communities in terms of the individual characteristics and socioeconomic circumstances that make people more susceptible to the impact of a hazardous event.

*Geographic Context*

The archipelago of Cape Verde is located in the Atlantic Ocean. It lies between latitudes 17°12′ and 14°48′ N and longitudes 22°44′ to 25°22′ W (Figure 1). There are 10 islands and 8 minor islets arrayed in a west-facing horseshoe. The islands are traditionally divided into the Barlavento (windward) group, comprising the islands of Santo Antão, São Vicente, Santa Luzia, São Nicolau, Sal, and Boa Vista, and the Sotavento (leeward) group, comprising Maio, Santiago, Fogo, and Brava. In terms of structure, the archipelago is located in a continental interpolate situation [31] whose origin is related to a hotspot mechanism (mantle plumes). The hotspot type activity would have begun around 19 to 22 million years ago and resulted in an uplift crustal area in which the archipelago is implanted [32], with volcanic activity still being a feature today. The islands are formed by alkaline basalts and their derived products, mainly post-Mesozoic formations that are predominantly mafic in nature and formed by sub-saturated silicates—especially basanites, tephrites, and nephelinites [33,34]. São Vicente has a diversified morphology with maximum altitudes up to 744 m in the Monte Verde and 395 m in Monte Topona. The city of Mindelo is surrounded by the slopes of what remains of the volcano that originated the island and also serves as a limit to the city. Rainfall is scant and highly variable. In general, rain falls in the form of showers, at times in heavy downpours that can reach values equal to or above monthly mean values. The rainy season is between August and October, and can sometimes start in July, associated with the Intertropical Convergence Zone [34,35].

The average monthly temperature varies between 22 °C in January and February and 27 °C in August and September. These are also the months when rainfall is higher. It occurs with great intensity. The annual precipitation is 51 mm.

São Vicente is the second most populous island in the archipelago. It represents about 15% of the population, with about 76,000 inhabitants and a population density close to 335 h/km$^2$. The most present economic activities in S. Vicente are in the tertiary sector, followed by the industrial sector, which presents a recent strong bet through investment in the Lazareto industrial [36].

São Vicente faces a number of natural hazards particularly flash flooding [37,38]. The report coordinated by the Sílvia Monteiro Survey of historical data on disasters in Cape Verde 1900–2013 identifies flash floods as phenomena that cause substantial economic losses.

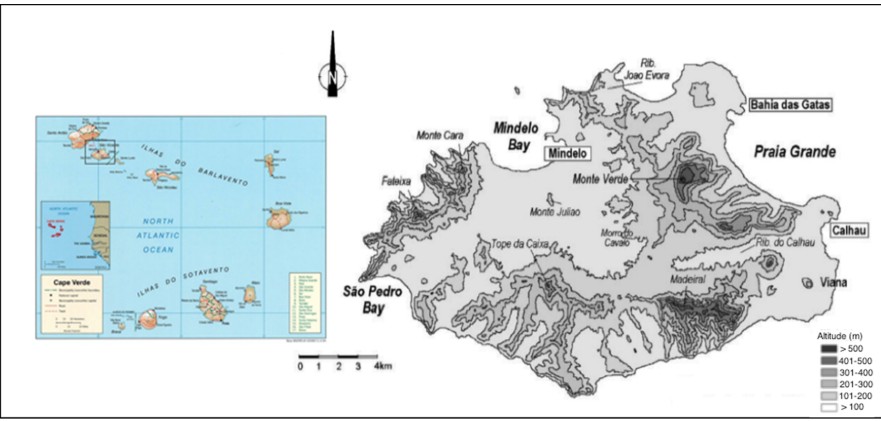

**Figure 1.** São Vicente and its location in the Cape Verde archipelago.

Periods of intense and concentrated precipitation contribute to a significant loss of soil and generate powerful torrents due to the steep slopes that are filled with small materials. The poor vegetation cover also contributes to creating particularly active dynamics for water courses. Of the islands, São Vicente has the third highest number of recorded events. In the period analyzed, 58 people were killed, 138 injured, 14 rescued, and 2000 evacuated in the archipelago. The report does not specify deaths in São Vicente, but it does identify a large number of displaced persons, suggesting the increasing frequency of this hazard. This increase is directly related to the rapid urban growth that has led to the construction of houses, some of which are illegal, as well as roads, which often occupy small river beds. These are dry for most of the year and sometimes for years at a time, but they nonetheless fill rapidly during more intense and prolonged rain. This accelerated construction process is one of the more important key factors in the heightened vulnerability in the face of flash floods [37,38].

From the point of view of preventing this hazard, measures have to be taken that delay the runoff's response to intense rain by increasing the time of concentration and therefore reducing the velocity of the surface runoff. However, the disorganized growth of the city has helped to destroy important drainage channels that were built to channel the surface waters and to increase the drainage speed so that it would reach the sea more quickly. Therefore, although the process or physical phenomenon has remained practically unchanged, an inadequate response strategy is significantly increasing the consequences of the hazard (Figure 2).

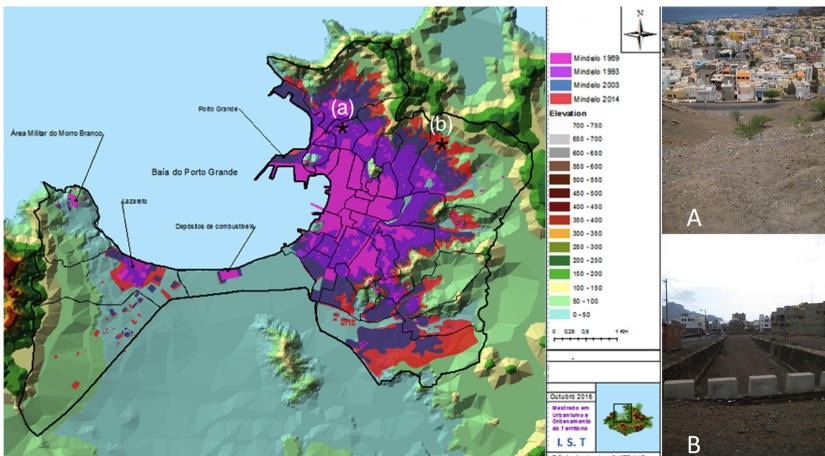

**Figure 2.** Mindelo city expansion evolution (1969, 1993, 2003, and 2014), adapted from Andrade (2017) [38]; (**A**)—Ditch whose alluvial cone coincides with the road that links Cruz João d' Évora to Morada in Madeiralzinho. (**B**)—Detritus flow resulting from heavy rainfall on 26 August 2008, affecting a very significant group of dwellings in Cruz São Évora (localization on map).

## 2. Materials and Methods

### 2.1. Sample and Questionnaire Characteristics

The theoretical framework of the study is based on the "psychometric paradigm" [39–41] which attempts to quantify individuals' risk perception through a survey questionnaire. A questionnaire entitled Flash Flood Hazard Perception in Cape Verde was developed for this study and randomly applied to 244 subjects resident in Mindelo, São Vicente, between April and May 2015. A total of 199 questionnaires were considered. This instrumental methodology is a well-established tool for natural hazard research. It is used to acquire information on participants' social characteristics, present and past behavior, standards of behavior or attitudes and their beliefs and reasons for action with respect to the topic under investigation [42]. It is also in line with Freixo (2009) [43], Pocinho (2012) [44], and Mendes (2015) [45], in which the need for a quantitative structuring of the field observed results are advocated, so as to define and systematize response configurations and patterns. Given the proposed goals, this questionnaire consisted of five parts (Table 1).

The first part characterizes the socio-economic and demographic factors of the population (gender, age, education, income), the type of residence and the prior experience of flash floods.

Based on the work by Burn (1999) [46], the second part of the questionnaire, "Perceptions regarding the flash flood hazard", has nine questions designed to evaluate how respondents perceive the risk of flash flood. The third part analyzes the causal attributions of flash floods, and the fourth assesses the support provided by government entities in case of crisis by the residents.

**Table 1.** Components of the questionnaire applied.

| | | |
|---|---|---|
| | Subjects' Characterization | (i) Gender<br>(ii) Age<br>(iii) Education level<br>(iv) Income<br>(v) Type of housing<br>(vi) Prior experience of flash floods |
| Flash Flood Risk Perception | Perceptions regarding the flash flood hazard | (i) Perceived as personal hazard<br>(ii) Perceived probability of dying as a result of exposure to a crisis event<br>(iii) The perceived degree of scientific knowledge about hazard<br>(iv) Knowledge of the hazard to which they are exposed (recent or old)<br>(v) The emotion of fear evoked by hazard in crisis situations<br>(vi) The possibilities of influencing hazard<br>(vii) The perceived frequency of crisis<br>(viii) Predictability<br>(ix) Increase or decrease of crisis events in the future |
| | Causal attributions of flash floods | (i) Fate<br>(ii) Unpredictable natural event<br>(iii) Divine punishment<br>(iv) Nature's vengeance<br>(v) Planning policies<br>(vi) Climate change |
| | Perception of support from government entities in case of crisis | (i) Support from local government<br>(ii) Support from central government |
| | Knowledge of flash flood hazard | (i) An emergency plan<br>(ii) The damage to which the house is exposed<br>(iii) The city's degree of exposure to the phenomenon of flash floods<br>(iv) The number of victims resulting from flash floods |

The last section of the questionnaire is "Knowledge of flash flood hazard" to which the subjects are exposed. The literature on hazards lets us conclude that the knowledge we have on hazards is positively related to the adoption of hazard reduction behavior [47,48]. A four-point Likert scale was used to measure residents' perceptions, ranging from 1, 'strongly disagree', at the lower end, which stands for not being important at all, with significance increasing along the scale up to 4, 'strongly agree', at the higher end, which indicates that a factor is a very important attribute [48]. The four-point Likert scale is also called a forced Likert scale since it avoids central tendency biases from the respondents and forces them to form an opinion, i.e., there is no safe 'neutral' option [49].

## 2.2. Statistical Analysis

Multiple statistical methods were applied to estimate the impact of particular factors on flash flood risk perception and to find relationships between awareness, causal attributions, and support from government entities. In most cases, these were correlation analysis [50–53], independent samples *t*-tests [54], one-way analysis of variance ANOVA [25,55], and regression analysis [30,56,57].

Quantitative statistics were primarily used to describe and summarize the features of the socio-demographic characteristics of the respondents and the other four impact factors (flash flood perception, causal knowledge, flood protection responsibility, trust in government). Analysis of variance (ANOVA) was then performed to examine the mean ranks of two or more independent variables with the null hypothesis of equality. Analysis of variance (ANOVA) was then used to identify whether or not there were any statistically significant differences between the impact factors and public flash flood risk perception. Next, post hoc tests were run to find and compare the flash flood risk perception differences between different groups under the same impact factor among all respondents. We used Tukey's B multiple comparison analysis to test each experimental group against each control group, since the Tukey method is preferred if there are unequal group sizes in the experimental and control groups. Moreover, with regard to the gender and prior experience variables, the independent *t*-test was used for the mean difference comparison between gender and prior experience because there were only two groups, male and female, and with and without experience.

Pearson correlations were also performed to assess possible collinearity among the independent variables. A correlation coefficient threshold between variables of $|r| > 0.7$ ($p < 0.05$) was considered an appropriate indicator for the point where collinearity begins to severely distort model estimation and subsequent prediction [57]. In the correlation matrix, there were few pairs of extremely correlated factors, underlining the high correlations among age, income per capita, and education.

All statistical analyses were carried out under a significance test value of 0.05 to confirm whether these impact factors affected the public flood risk perception. Impact factors with a significance of less than 0.05 were considered to significantly influence public flood risk perception, as proposed in similar questionnaire-based studies [30,55,57].

The data were analyzed using the IBM SPSS software (Version 24, SPSS Inc., Armonk, NY, USA).

## 3. Results

### 3.1. Characteristics of the Respondents

The study sample comprised 199 subjects residing in Mindelo, S. Vicente island, Cape Verde. Of the respondents, 37.2% (74 individuals) were male and 62.8% (125 individuals) are female. The age of the respondents ranged from 17 to 72 years (with an average of 35 years). Most of the respondents claimed to have secondary or higher education (28.1% and 29.1%, respectively). Those who could not read and answer amounted to 6%, and 26.1% had the first and second school cycles, while 10.6% had the third school cycle. In terms of income, 22.1% of respondents earn between 45.36 and 90.71 euros, 18.6% claim not to have any income, 10.1% claim to earn between 453.57 and 907.13 euros, and 9% claim to have an income less than 45.36 euros. Around 7 subjects (3.5%) did not answer this question. When asked about the type of house, 42.2% (84 respondents) mentioned owning a house. Of the

surveyed residents, 22.1% (44 respondents) mention living in a rented house and 14.6% claim to be in another situation. Around 42 subjects (21.1%) did not answer this question. A majority of respondents (73.4%) mention having had past experience of the flash flood hazard, against 23.1% (46 people) who say they have had no previous direct contact with the phenomenon of flash floods. Around 3% of the subjects did not answer this question.

Based on Pearson correlation matrix, there were some pairs of highly correlated factors, underscoring the high correlations between age, per capita income and education. Therefore, four independent variables were selected for the statistical analyses: gender, level of education, type of house, and prior experience.

### 3.2. Public Flash Floods Risk Perception

Considering the average values regarding the perceptions in the face of hazard of flash floods, it was found that the subjects agreed (albeit partially) that the flash floods in S. Vicente: (i) are a high personal hazard; (ii) are a fatality; (iii) provoke fear; (iv) are not predictable; (v) will tend to increase in the future. However, the subjects disagreed as to whether flash floods are a phenomenon: (i) known by science; (ii) old; (iii) possibly influenced by human action relative to their occurrence; (iv) whose occurrence is rare. As for the causal attributions in the face of flash flood hazard, the average values found indicate that the respondents agreed that flash floods are an unpredictable natural event and result from climate change. Nevertheless, the people surveyed disagreed about whether flash floods are twist of fate, a divine punishment, nature's vengeance, or result from inadequate planning policies. Concerning the Perception of support from government entities in the case of flash floods, the people surveyed believe they receive insufficient support from the local and central government. Regarding the knowledge of the flash flood hazard, the subjects disagreed about whether there is an adequate emergency plan and whether their own house is liable to damage; however, they agreed that the city is liable to damage and that loss of life can occur.

### 3.3. Impact Factors of Public Flash Flood Risk Perception

A one-way ANOVA analysis was performed to examine whether the flood risk perception in Mindelo differed across the analyzed groups. The ANOVA results showed that the independent factors of gender, education level, type of house, and prior experience have significant relationships with the flash flood risk perception and statistically significant differences were identified between groups (Table 2). Gender has statistically significant differences (*p*-value < 0.05) in 11 variables, and the level of education, the type of housing, and prior experience influence 16 variables related to flash flood risk perception differently. The role of central and local government is the variable that denotes the greatest statistical difference among the factors analyzed. The *t*-test was conducted to examine the differences in risk perception between the two genders for the variables identified by ANOVA as showing a statistically significant difference. The average results showed that there is significant difference in risk perception between males and females, both for Levene's test for equality of variances and for the *t*-test for equality of means as the significant values were lower than the significant value of 0.05. Comparison showed that the mean score for females (2.7172) was lower than that for males (2.8182), as shown in Table 3. This result suggests that males had higher risk perception than females. Although it is assumed that the males perceive more risk than the female respondents, females seem to be more sensitive to several components of the risk perception, such as: provokes fear, twist of fate, unpredictable natural event, and city liable to damage. Conversely, males consider the type of risk to be a twist of fate, an old hazard that occurs seldom. Males also show more confidence in receiving support from local and central government (Table 3).

**Table 2.** ANOVA tests for public flash risk perception against different socio-demographic factors.

| Perceived as | Gender | | Education | | Type of Housing | | Prior Experience | |
|---|---|---|---|---|---|---|---|---|
| | F | Sig. | F | Sig. | F | Sig. | F | Sig. |
| Personal hazard | 2.683 | 0.103 | 3.150 | **0.015** | 31.159 | **0.000** | 4.436 | **0.013** |
| Probably not fatal | 13.344 | **0.000** | 0.480 | 0.750 | 1.028 | 0.312 | 2.854 | 0.060 |
| Known by science | 5.615 | **0.019** | 33.277 | **0.000** | 9.082 | **0.003** | 10.584 | **0.000** |
| Ancient hazard | 17.078 | **0.000** | 18.977 | **0.000** | 14.556 | **0.000** | 10.767 | **0.000** |
| Provokes fear | 21.177 | **0.000** | 1.782 | 0.134 | 0.567 | 0.452 | 2.817 | 0.062 |
| Possible influence | 0.521 | 0.471 | 9.942 | **0.000** | 7.623 | **0.006** | 3.379 | **0.036** |
| Seldom occurs | 8.059 | **0.005** | 9.754 | **0.000** | 6.693 | **0.010** | 6.177 | **0.002** |
| Predictable | 0.515 | 0.474 | 233.813 | **0.000** | 19.766 | **0.000** | 21.541 | **0.000** |
| Positive trend | 0.982 | 0.322 | 63.275 | **0.000** | 13.975 | **0.000** | 26.165 | **0.000** |
| *Causal factors* | | | | | | | | |
| Twist of fate | 12.260 | **0.001** | 1.273 | 0.282 | 2.215 | 0.112 | 0.002 | 0.962 |
| Unpredictable natural event | 23.464 | **0.000** | 1.829 | 0.125 | 2.506 | 0.132 | 0.248 | 0.619 |
| Divine punishment | 1.784 | 0.183 | 75.248 | **0.000** | 19.497 | **0.000** | 39.290 | **0.000** |
| Nature's vengeance | 1.026 | 0.312 | 30.907 | **0.000** | 9.535 | **0.000** | 19.947 | **0.000** |
| Planning policies | 3.591 | 0.060 | 62.894 | **0.000** | 23.950 | **0.000** | 35.960 | **0.000** |
| Climate change | 3.849 | 0.051 | 1.386 | 0.240 | 0.396 | 0.673 | 0.125 | 0.724 |
| *Support from government* | | | | | | | | |
| Local government | 19.096 | **0.000** | 16.617 | **0.000** | 15.775 | **0.000** | 11.510 | **0.001** |
| Central government | 29.808 | **0.000** | 27.865 | **0.000** | 11.494 | **0.000** | 10.287 | **0.002** |
| *Damage* | | | | | | | | |
| Proper emergency plan | 41.204 | **0.000** | 29.422 | **0.000** | 13.869 | **0.000** | 10.816 | **0.001** |
| House liable to damage | 0.019 | 0.892 | 4.596 | **0.001** | 5.535 | **0.005** | 7.770 | **0.006** |
| City liable to damage | 6.150 | **0.014** | 7.005 | **0.000** | 11.110 | **0.000** | 90.597 | **0.000** |
| Loss of life | 2.677 | 0.103 | 7.424 | **0.000** | 10.374 | **0.000** | 74.972 | **0.000** |

Note: The bold numbers indicate statistically significant differences between groups ($p$-value $< 0.05$); n = 199.

**Table 3.** Mean, variance, Levene's, and $t$-test for risk perception by gender.

| Gender | Mean | | Variance | | Levene's Test for Equality of Variances | | $t$-Test for Equality of Means | |
|---|---|---|---|---|---|---|---|---|
| | Male | Female | Male | Female | F | Sig. | $t$ | Sig. (2-Tailed) |
| Probably not fatal | **3.6133** | 3.3095 | 0.240 | 0.375 | 3200 | 0.075 | 3.653 | 0.000 |
| Known by science | 2.3867 | **2.7302** | 0.916 | 1.031 | 1345 | 0.248 | −2.369 | 0.019 |
| Ancient hazard | **2.9600** | 2.3175 | 0.958 | 1.242 | 6.168 | 0.014 | 4.133 | 0.000 |
| Provokes fear | 3.0667 | **3.4762** | 0.441 | 0.331 | 2.080 | 0.151 | −4.602 | 0.000 |
| Seldom occurs | **2.7333** | 2.3492 | 0.874 | 0.853 | 0.031 | 0.862 | 2.839 | 0.000 |
| Twist of fate | 2.0800 | **2.5159** | 0.696 | 0.748 | 2.202 | 0.139 | −3.501 | 0.000 |
| Unpredictable natural event | 2.8267 | **3.3175** | 0.578 | 0.426 | 0.316 | 0.575 | −4.844 | 0.000 |
| Local government support | **2.7867** | 2.3254 | 0.630 | 0.461 | 0.551 | 0.459 | 4.370 | 0.000 |
| Central government support | **2.7200** | 2.1429 | 0.421 | 0.587 | 3.325 | 0.070 | 5.460 | 0.000 |
| Proper emergency plan | **2.8267** | 2.1429 | 0.524 | 0.539 | 0.023 | 0.881 | 6.419 | 0.000 |
| City Damage | 3.0000 | **3.2619** | 0.486 | 0.538 | 5.949 | 0.016 | −2.517 | 0.013 |
| **Average** | **2.8182** | 2.7172 | 0.744 | 0.893 | 24.943 | 0.000 | 2.569 | 0.010 |

With regard to the importance of prior experience the $t$-test also demonstrates differences in flash flood risk perception between the respondents with and without experience. Whereas Levene's test for equal variances is not statistically significant, since the $p$-value is higher than 0.05, the $t$-test for equality of means is significant, being lower than value of 0.05. The mean score for residents with prior experience (2.7048) was found to be higher than for those without experience (2.2351), as shown in Table 4. This result suggests that residents who had experienced this type of risk had a higher risk perception than residents who had never encountered the consequences of a flash flood.

**Table 4.** Mean, variance, Levene's, and *t*-test for risk perception by prior experience.

| Prior Experience | Mean | | Variance | | Levene's Test for Equality of Variances | | *t*-Test for Equality of Means | |
|---|---|---|---|---|---|---|---|---|
| | With | Without | With | Without | F | Sig. | t | Sig. (2-Tailed) |
| Personal hazard | **3.2903** | 2.6304 | 0.558 | .283 | 5.291 | 0.022 | 5.582 | 0.000 |
| Known by science | **2.7161** | 2.2174 | 0.958 | 1.018 | 0.439 | 0.508 | 3.014 | 0.003 |
| Ancient hazard | 2.4000 | **3.0870** | 1.203 | 0.970 | 2.588 | 0.109 | −3.815 | 0.000 |
| Possible influence | **2.4258** | 2.0000 | 0.856 | 0.800 | 0.766 | 0.383 | 2.761 | 0.006 |
| Seldom occurs | 2.4000 | **2.8043** | 0.904 | 0.739 | 1.725 | 0.191 | −2.587 | 0.010 |
| Predictable | **2.5871** | 1.8043 | 1.153 | 0.916 | 2.075 | 0.151 | 4.446 | 0.000 |
| Positive trend | **2.7484** | 2.1957 | 0.449 | 0.294 | 5.066 | 0.025 | 5.115 | 0.000 |
| Divine punishment | **2.4323** | 1.4783 | 0.935 | 0.433 | 11.379 | 0.001 | 7.677 | 0.000 |
| Nature's vengeance | **2.5677** | 1.9130 | 0.753 | 0.792 | 0.406 | 0.525 | 4.466 | 0.000 |
| Planning policies | **2.4839** | 1.7826 | 0.511 | 0.396 | 4.222 | 0.041 | 6.426 | 0.000 |
| Local government support | **2.5935** | 2.1739 | 0.503 | 0.680 | 0.066 | 0.798 | 3.393 | 0.001 |
| Central government support | **2.4516** | 2.0435 | 0.561 | 0.620 | 0.275 | 0.600 | 3.207 | 0.002 |
| Proper emergency plan | **2.4968** | 2.0652 | 0.615 | 0.596 | 2.412 | 0.122 | 3.289 | 0.001 |
| House liable to damage | **2.7419** | 2.5000 | 0.271 | 0.256 | 2.160 | 0.143 | 2.787 | 0.006 |
| City liable to damage | **3.3871** | 2.4130 | 0.395 | 0.292 | 2.103 | 0.149 | 9.518 | 0.000 |
| Loss of life | **3.5548** | 2.6522 | 0.404 | 0.321 | 2.233 | 0.137 | 8.659 | 0.000 |
| Average | **2.7048** | 2.2351 | 0.815 | 0.743 | 3.713 | 0.054 | 12.525 | 0.000 |

In order to understand which groups are significant different in terms of level of education and type of housing, we further conduct Tukey's B analysis, a post hoc test, to obtain detailed results. Table 5 shows the results for the different risk perceptions of respondents with different levels of education. In general, the group with the best education has the highest risk perception (3.0488) among the four groups. Conversely, the illiterate respondents who cannot read or write and those who have the first and second cycle recorded the lowest values of risk perception. However, a detailed analysis of the results shows that it is the respondents with high levels of education who believe that flash flooding is due to divine punishment and nature's vengeance.

**Table 5.** Tukey's B results for different educational groups.

| Education | CRW | First/Second Cycle | Third Cycle | Sec. Edu. | Hig-Edu. |
|---|---|---|---|---|---|
| Personal hazard | 2.6667 [a] | 2.9855 [ab] | 3.2273 [b] | **3.3333 [b]** | 3.2195 [b] |
| Known by science | 1.8333 [a] | 1.8986 [a] | 2.5000 [b] | 3.0526 [c] | **3.4390 [c]** |
| Ancient hazard | **3.2500 [b]** | 3.1884 [b] | 2.7727 [b] | 2.0877 [a] | 1.8293 [a] |
| Possible influence | 2.2500 [ab] | 1.8261 [a] | 2.4091 [ab] | **2.7018 [b]** | 2.6341 [b] |
| Seldom occurs | **3.1667 [c]** | 2.8551 [c] | 2.6818 [bc] | 2.1404 [ab] | 2.0732 [a] |
| Predictable | 1.1667 [a] | 1.4058 [a] | 2.0000 [b] | 2.9825 [c] | **3.8780 [d]** |
| Positive trend | 1.9167 [a] | 2.0870 [a] | 2.5000 [b] | 2.9649 [c] | **3.3171 [d]** |
| Divine punishment | 1.3333 [a] | 1.3768 [a] | 1.9091 [b] | 2.8772 [c] | **3.1220 [c]** |
| Nature's vengeance | 1.6667 [a] | 1.8551 [a] | 2.0455 [a] | 2.9123 [b] | **3.0976 [b]** |
| Planning policies | 1.7500 [a] | 1.6957 [a] | 2.1364 [b] | 2.7544 [c] | **3.0488 [c]** |
| Local government support | 2.3333 [a] | 2.2273 [a] | 2.0725 [a] | 2.8246 [b] | **2.9512 [b]** |
| Central government support | 1.5000 [a] | 1.9130 [b] | 2.2273 [b] | 2.7193 [c] | **2.9268 [c]** |
| Proper emergency plan | 1.7500 [a] | 1.8841 [a] | 2.1818 [a] | 2.8421 [b] | **2.9512 [b]** |
| House liable to damage | 2.4167 [a] | 2.5217 [ab] | 2.7273 [ab] | 2.8049 [b] | **2.8421 [b]** |
| City liable to damage | 3.0000 [ab] | 2.8406 [a] | 3.1818 [ab] | 3.3860 [b] | **3.4390 [b]** |
| Loss of life | 3.0833 [a] | 3.0290 [a] | 3.4091 [ab] | 3.5614 [b] | **3.6341 [b]** |
| Average | **2.1889 [a]** | **2.2406 [a]** | **2.5152 [b]** | **2.8854 [c]** | **3.0488 [d]** |

Value within each group followed by different letters are significantly different at *p* ≤ 0.05. Means for groups in homogeneous subsets are displayed. (i) Uses Harmonic Mean Sample Size = 26.996. (ii) The group sizes are unequal. The harmonic mean of the group sizes is used.

Tukey's B analysis was also run to define two main groups according to housing. The two groups with the highest values were the respondents who are homeowners and those who rent their homes. This gives a better perception of flash flood risk (Table 6).

**Table 6.** Tukey's B results for different type of housing.

| Type of House | Home Ownership | Rented | Other Situation |
|---|---|---|---|
| Personal hazard | **3.2167** [b] | 3.1957 [b] | 2.8000 [a] |
| Ancient hazard | 2.3083 [a] | 2.6957 [a] | **3.2286** [b] |
| Known by science | **2.7917** [b] | 2.6087 [b] | 1.9429 [a] |
| Possible influence | 2.3750 [b] | **2.4783** [b] | 1.9714 [a] |
| Seldom occurs | 2.3500 [a] | **2.5000** [a] | 2.9714 [b] |
| Predictable | **2.7417** [c] | 2.2174 [b] | 1.5143 [a] |
| Future trend | **2.7667** [b] | 2.6304 [b] | 2.1143 [a] |
| Divine punishment | **2.4833** [b] | 2.1304 [b] | 1.4000 [a] |
| Nature's vengeance | **2.5917** [b] | 2.3913 [b] | 1.8571 [a] |
| Planning policies | **2.5333** [b] | 2.3043 [b] | 1.6286 [a] |
| Local government support | **2.6750** [b] | 2.4783 [b] | 1.9143 [a] |
| Central government support | **2.5417** [b] | 2.2391 [b] | 1.8857 [a] |
| Proper emergency plan | **2.6000** [b] | 2.2826 [b] | 1.8571 [a] |
| House liable to damage | **2.7750** [b] | 2.6304 [ab] | 2.4571 [a] |
| City liable to damage | **3.3167** [b] | 3.1304 [b] | 2.6857 [a] |
| Loss of life | **3.5083** [b] | 3.2609 [b] | 2.9143 [a] |
| **Average** | **2.7234** [b] | **2.5734** [b] | **2.1964** [a] |

Value within each group followed by different letters are significantly different at $p \leq 0.05$. Means for groups in homogeneous subsets are displayed. (i) Uses Harmonic Mean Sample Size = 51.156. (ii) The group sizes are unequal. The harmonic mean of the group sizes is used.

## 4. Discussion

Several authors [12,15,25,57–60] consider that in almost every study on flood risk perception in which socio-demographic characteristics are examined the most important characteristics seem to be age, gender, education level, income, home ownership, as well as direct experience of floods in previous years. Our study found that age and income are highly related to education (r: −0.755; r: 0.824, respectively), meaning that it is the younger people and those with higher education levels who recorded the highest level of income. In fact, regarding the level of education, higher-educated respondents show a better flash flood risk perception [25,61–64]. Ho et al. (2009) [62] suggest that people with more years of education may acquire and understand new information more easily. As a result, they may be aware of more mitigation actions implemented by local governments and experts and thus may feel that a higher degree of control can be exercised over a disaster. Lopez-Marrero and Yarnal (2010) [63] report a positive association between income and housing conditions (construction materials) and housing location, as people with lower incomes will predominantly live in poorer housing conditions in less favorable areas (e.g., flood-prone areas). Shah et al. (2018) [64] when analyzing the household vulnerability and resilience to flood hazards in disaster-prone in various districts of Pakistan consider that literacy is an important component of the household sensitivity to flood hazards: less-educated households are less likely to adopt advance coping strategies and are more sensitive to floods due to lack of forward-looking behavior. Conversely, several other authors [57,59,64,65] found no or a negative relationship between perceived risk and education level and income.

A significant number of studies highlight the role of individual experiences in flood risk perception [65]. Significant statistical differences between the groups with and without direct experience of flash floods have been found concerning the perceptions, causal attributions, knowledge, and perception of support from public entities regarding flash flood hazard. People with direct experience of flash flooding had risk perception and perceived the hazard to be more frequent; they also

see themselves as future victims, which increases the motivation to engage in hazard reduction behaviors [10,28,46,66–69].

Gender is strongly related to risk judgments and attitudes [39]. Several studies [55–57,63] found that, on average, males have lower flood risk perception levels than females, although Botzen (2009) [58] suggests an opposite relation. The results obtained in this study suggest that men have higher levels of knowledge, which could be related to biological, social, and cultural factors; women are found to have a high level of fear of flash flood risk since in their particular type of society they are characterized as physically more vulnerable [70] or are more concerned about human health [58].

The literature regarding risks suggests that owning a house is associated with a higher level of risk perception than renting a house [50,68,71]. Our study found no significant differences between these groups.

In our study, the poorer perception seems to be related to the causal attributions regarding the flash flood risk, mainly involving an external cause such as divine punishment or nature's vengeance. For several authors [72–74], hazards and risks are socially constructed, i.e., connected with the dynamics of the social system—culture, institutions, values, beliefs, etc.—in a complex manner. According to Holmgaard (2019) [75], religious beliefs can shape how people perceive disaster risk, how they respond to disasters and recover from their impact. In the Cape Verde Islands, as in other similar societies in developing countries, most people have local spiritual beliefs that pervade their everyday activities and are commonly used to rationalize various incidents and observations [76–79]. In this context, the problem is especially that if a person believes a phenomenon has a divine cause, then not only is it useless to try and change it, but it is also an offence to the divinity to contemplate doing so [78].

Because risk perception does not occur in a social vacuum one cannot account for how people perceive and understand risks without also considering the social contexts [79]. In this respect, Douglas (1978) [80] considers that risk perception is a socially, or culturally, constructed phenomenon, that is, however, also governed by personality traits, needs, preferences, or properties of the risk objects. Both perspectives need to be brought into the formal decision-making process. What is perceived as dangerous, and how much risk to accept, are a function of a person's cultural adherence and social learning. Therefore, strategies for eliciting optimal responses to such risk emphasize effective risk communication grounded in an appropriate sociocultural context. Consequently, knowledge about which aspects or characteristics of the risk source are important for subjective risk judgements may influence such demands, and hence also political actions aimed at reducing the risk.

## 5. Conclusions and Further Research

The present study has at least three important findings. First, risk perception represents a combination of science and judgment with significant social, cultural, and political interpretations. As a matter of fact, the subjects agreed that the flash floods are a high personal hazard, provoke fear, are not predictable and will tend to increase in the future. Although it is known by science, respondents agreed that flash floods are an unpredictable natural event and are related with climate change. They agreed that the city is liable to damages and that loss of life can occur. Nevertheless, the people surveyed disagreed about whether flash floods are a divine punishment or consequence of inadequate planning policies, since they believe they receive insufficient support from the local and central government.

Second, risk perception is intrinsically subjective, related with the age, gender, education, type of house, and prior experiences. In this sense, flash flood risk perception is a complex process that encompasses both cognitive (e.g., knowledge, likelihood, etc.) and affective (e.g., feelings, beliefs, perceived control, etc.) features, wherein local conditions have a major effect on people's knowledge and behavior. In face of these results, studies on the factors of flood risk cannot be limited only to metrical (statistical) data, they should also include people's social and cultural interpretation of the flood risk. Risk perception needs to be approached in a more holistic manner, embracing contextual factors such as culture, religion, history, and political contexts. The conceptions of vulnerability,

capacity of response, and resilience should also be analyzed in an integrated model of perceived risk and risk-reducing strategies in order to reduce the ambiguity and complexity of risk perception.

Lastly, we believe that the results presented here can help flood managers in their development of national and local flood risk management strategies, as well as preparing risk communication, which necessarily involves the complexity of individual/cultural risk perceptions. Defining clear strategies in people's flood-risk behavior is a priority. When it comes to creating educational programs on flash-flood risks, which could be accomplished through training sessions, presentations at public functions, informational fliers, and other sources of communication, the focus must be on understanding the causes and possible consequences of floods, on increasing awareness of warning signals, and on informing the public about available tools and data.

**Author Contributions:** All authors contributed to conceiving and designing the article: conceptualization, B.M. and A.N.; methodology, B.M.; A.N.; L.L. and F.V.-C.; investigation B.M; A.N.; L.L. and F.V.-C.; writing—review and editing, B.M.; A.N.; L.L. and F.V.-C.

**Funding:** This work was supported by the European Regional Development Funds, through the COMPETE 2020—Operational Programme 'Competitiveness and Internationalization', under grant no. POCI-01-0145-FEDER-006891; and by National Funds through the Portuguese Foundation for Science and Technology (FCT) under Grant UID/GEO/04084/2013.

**Conflicts of Interest:** The authors declare no conflict of interest.

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
