# Peer review of "Flash Flood Risk Perception by the Population of Mindelo, S. Vicente (Cape Verde)"

_water, doi:10.3390/w11091895_

Round 1

Reviewer 1 Report

I read with interest the manuscript entitled "Flash flood risk perception by the population of Mindelo, S. Vicente (Cape Verde)". The authors conducted a survey to shed light on the factors affecting flood risk perception.

In my opinion this paper presents an interesting study. Understanding the local context is essential to plan adequate flood management strategies. The manuscript is well written and the results are analysed into detail. I would like to recommend the publication of this manuscript after major revisions.

Firstly, I would like to recommend re-organising a bit the results and discussion to better highlight the take home messages of the paper.

Second, I was wondering whether an in-depth comparison between the results of this study and previously published literature could be added to the section "Discussion".  I think that such a discussion would increase the scientific value of the manuscript and help to provide a more comprehensive understanding of the problem at the global scale.

Third, I was wondering whether the authors explored the links between the religious beliefs and perceptions and the demand for support from government entities in case of crisis. How do lines 366-367 relate to the demand for support from the public institutions?

Finally, did the authors think to investigate which mitigation strategy could be most likely accepted by the population? The answer to this latter question depends on the factors investigated by the authors and it is pivotal for improved flood management, risk mitigation, loss and casualties reduction.

I hope the authors will find my comments and questions useful to improve their manuscript.

Please find below my minor comments.

Line 38: please correct centurys”.

Line 45: please remove the full stop after the brackets.

Line 90: please clarify the meaning of “when this is more to the north”.

Figure 1; the inset on the top left showing the location of the study area is a bit too small. The words are hard to ready. Would it be possible to increase size and resolution of this inset?

Line 114-115: I suggest making the subject explicit: “Rainfall occurs with high intensity”.

Line 116: please add a comma: “natural hazards, particularly…”

Line 118: I think it would be useful to add some quantitative evaluation. For instance, the impact of floods on GDP.

Lines 119-121: this is a long sentence without punctuation. I suggest splitting this sentence into 2-3 statements to enhance the readability of the text.

Lines 186-187: To my understanding, Perceptions regarding the flash flood hazard is the title of the second section of the questionnaire. If so, I suggest writing the title in italics or in between inverted commas.

Line 190: Similarly to my previous comment, is Knowledge of [the] flash flood hazard the title of the last section of the questionnaire? If so, please make sure that this is clear to the reader.

Lines 191-192: please add a reference to support this statement.

Line 199: please provide the appropriate caption for Table 1

Line 306: please correct significantly.

Line 361: could please the authors clarify what they mean with socially constructed?

Author Response

The authors are gratefully for all the comments/suggestions about the paper. All the changes made in the paper are in yellow colour.

(1)    A slight re-organization was carried out with the aim of highlighting the most relevant points of the paper, especially regarding the conclusions, which were revised.

(2)     Some references were added to the item "discussion" in order to integrate the results obtained in broader contexts. However, it should be emphasized that the geo-human specificity of the study area joined with the methodology used does not facilitate the comparison with other results obtained in different geographic contexts and with varied methodological procedures.

(3/4) Despite being interesting topics, in this investigation, we were not been able to explore the links between the religious beliefs and perceptions and the demand for support from government entities in case of crisis. Similarly, it was not investigated which or what the best mitigation strategies, post-event. As we mentioned in the conclusion this kind of topic needs to be approached in a more holistic manner, embracing contextual factors such as culture, religion, history political contexts, national and local flood risk management strategies.  

Regarding the minor comments, all the amendments were made accordingly. 

Reviewer 2 Report

The manuscript titled "Flash flood risk perception by the population of Mindelo, S. Vicente (Cape Verde)" presents the results of a field study, regarding the flash flood risk, at Cape Verde. The research is interesting and innovative, whereas discussion and conclusion are in line with the results. The manuscript is well written. Thereofre I suggest to be published in the current form.

Author Response

The authors deeply thank the reviewer for carefully reading the paper.

Reviewer 3 Report

The manuscript entitled “Flash flood risk perception by the population of Mindelo, S. Vicente (Cape Verde)” is very interesting dealing with the identification of the primary factors associated with the perception of flash flood risk. In general the structure of the paper is good and even if I am not a native speaker I have to say that it is written in good English language.

Here are some comments and suggestions that could improve the quality of the final version of the paper:

A better location map is required showing the broader area of the country. The map of the island should depict also the drainage networks.

In line 115 the authors mention that “The average annual precipitation is 51 mm”. Usually the annual rainfall is given as the sum of the precipitation of all months and not as a average. Please give the annual rainfall of the study area.

In the Geographic Context section a description of the drainage networks of the island would be good for the reader. Additionally, the authors should provide a list with the flash flood events that have affected the study area (at least date of each event and mostly affected locations and amount of rain – if available).

In Fig 2 a map showing the locations of the photos would help the reader.

In the introduction section demographic data (population, population density, age of the inhabitants, and education level of the population) and data about the economic activity of the study area are necessary.

Author Response

The authors sincerely thank the suggestions made.

Figure 1 has been edited.
A map of Mindelo city expansion evolution (1969, 1993, 2003 and 2014) and the location of the photographs was added.
The authors added some information about São Vicente demographics features.
It was not possible to present a drainage networks map as suggested by the reviewer. Certainly will improve the work. However for its elaboration would implies work field, which is not possible in a short time. I hope the reviewer understands this impossibility.

Round 2

Reviewer 1 Report

I would like to thank you the authors for reading my comments and providing an answer to each one of my questions.

In my opinion, this paper is an interesting contribution to the literature.